# Gene Therapy for Neuronopathic Mucopolysaccharidoses: State of the Art

**DOI:** 10.3390/ijms22179200

**Published:** 2021-08-25

**Authors:** María José de Castro, Mireia del Toro, Roberto Giugliani, María Luz Couce

**Affiliations:** 1Unit of Diagnosis and Treatment of Congenital Metabolic Diseases, Department of Paediatrics, Santiago de Compostela University Clinical Hospital, 15704 Santiago de Compostela, Spain; mj.decastrol@gmail.com; 2IDIS, Health Research Institute of Santiago de Compostela, 15704 Santiago de Compostela, Spain; 3CIBERER, Centro de Investigación Biomédica en Red de Enfermedades Raras, 28029 Madrid, Spain; mdeltoro@vhebron.net; 4MetabERN, European Reference Network for Hereditary Metabolic Disorders, 33100 Udine, Italy; 5Pediatric Neurology Department, University Hospital Vall d’Hebron, Universitat Autónoma de Barcelona, CIBERER, MetabERN, 08035 Barcelona, Spain; 6Medical Genetics Service, Gene Therapy Center, Medical Genetics Clinical Research Group, Biodiscovery Research Group, HCPA, Porto Alegre 90035-903, Brazil; rgiugliani@hcpa.edu.br; 7Department of Genetics, UFRGS, Porto Alegre 91501-970, Brazil; 8DASA/GeneOne, São Paulo 04078-013, Brazil

**Keywords:** mucopolysaccharidoses, gene therapy, viral vectors, adeno-associated virus, lentivirus, central nervous system, blood brain barrier

## Abstract

The need for long-lasting and transformative therapies for mucopolysaccharidoses (MPS) cannot be understated. Currently, many forms of MPS lack a specific treatment and in other cases available therapies, such as enzyme replacement therapy (ERT), do not reach important areas such as the central nervous system (CNS). The advent of newborn screening procedures represents a major step forward in early identification and treatment of individuals with MPS. However, the treatment of brain disease in neuronopathic MPS has been a major challenge to date, mainly because the blood brain barrier (BBB) prevents penetration of the brain by large molecules, including enzymes. Over the last years several novel experimental therapies for neuronopathic MPS have been investigated. Gene therapy and gene editing constitute potentially curative treatments. However, despite recent progress in the field, several considerations should be taken into account. This review focuses on the state of the art of in vivo and ex vivo gene therapy-based approaches targeting the CNS in neuronopathic MPS, discusses clinical trials conducted to date, and provides a vision for the future implications of these therapies for the medical community. Recent advances in the field, as well as limitations relating to efficacy, potential toxicity, and immunogenicity, are also discussed.

## 1. Introduction

Mucopolysaccharidoses (MPS) is a group of rare, inherited lysosomal disorders caused by a deficiency in one of the eleven enzymes involved in the degradation of glycosaminoglycans (GAGs). The progressive accumulation of GAGs in the lysosomes leads to cellular dysfunction and cell death, resulting in multisystem clinical manifestations. Somatic involvement is heterogeneous and can include coarse facial features, organomegaly, bone and joint abnormalities, impaired vision, hearing loss, and cardiorespiratory disease [1,2,3]. In neuronopathic MPS, which includes MPS I (Hurler syndrome), MPS II (Hunter syndrome), MPS III (Sanfilippo A, B, C and D syndromes), and MPS VII (Sly syndrome), patients develop cognitive impairment, behavioral disturbances [4,5], sleep disorders [6] and epilepsy. In the final stage of the disease, they lose previously acquired skills resulting in a vegetative state. Brain changes associated with MPS disorders include white and gray matter alterations, ventriculomegaly, hydrocephalus, cortical atrophy, and enlargement of the perivascular spaces [7]. The pathophysiology of neuronal damage in neuronopathic MPS is mainly caused by the accumulation of heparan sulfate (HS) and secondary toxic products such as GM2 and GM3 gangliosides, inflammatory cytokines, and reactive oxygen species [8].

Current therapies for MPS predominantly seek to supply functional enzymes in order to reduce substrate accumulation in the CNS. These strategies rely on the mechanism of cross correction, whereby a functional enzyme secreted by wild-type cells is taken up and sorted to the lysosome via the mannose-6-phosphate receptor (M6P-R) pathway, restoring GAG metabolism. Currently, intravenous (IV) enzyme replacement therapy (ERT) and allogeneic hematopoietic stem cell transplantation (HSCT) are the standard treatments for the majority of MPS diseases [9,10]. To date, IV ERT has been approved for MPS I, MPS II, MPS IVA, MPS VI and MPS VII [7], and constitutes an effective treatment for several systemic clinical signs. However, the blood brain barrier (BBB) prevents penetration of the brain parenchyma by large molecules, including enzymes [11,12]. One strategy to provide enzyme replacement across the BBB involves the intrathecal or intracerebroventricular infusions, which are invasive procedures. Another strategy is the IV administration of fusion proteins, that combine the therapeutic enzyme with an antibody that targets specific receptors in the BBB (e.g., insulin and transferrin receptors), which allow the molecule to travel across the BBB [13]. Another approach to provide therapy that reaches the CNS is to perform HSCT, that provide donor stem cells that enter the blood as monocytes and differentiate into macrophages and microglia, which can cross the BBB and secrete the deficient enzyme [14]. HSCT is a standard and effective therapy for MPS I patients who present CNS involvement [15]. While data on its effects in other neuronopathic forms of MPS are limited, it may have an effect on CNS disease in certain patients with MPS II when administered in early disease stages [16,17].

Several different strategies that target the brain to treat CNS disease in MPS are being studied in preclinical [18,19,20] and clinical [21] settings. Given that MPS are monogenic hereditary disorders and that delivery of a small amount of therapeutic enzyme is sufficient to improve the associated somatic and CNS signs, these diseases are considered suitable candidates for gene therapy and gene editing approaches. Two main gene therapy strategies are described (Figure 1). In vivo gene therapy involves the direct introduction (into the blood stream, or into the affected organ) of a vector carrying the therapeutic gene into the patient [22]. This approach has proven successful in targeting the CNS, and has been approved by the FDA and the EMA for the treatment of inherited retinal disease [23] and spinal muscular atrophy (SMA) [24]. Ex vivo gene therapy consists of culturing the patient’s cells in the laboratory, transducing the cells with a viral vector carrying the desired gene, and then infusing the cells back into the patient [25]. This strategy has also been successful in treating CNS pathology, and the regulatory agencies recently recommended granting marketing authorization for two lentiviral haemopoietic stem cell gene therapy drugs to treat metachromatic leukodystrophy and adrenoleukodystrophy [26].

In general, all gene transfer approaches can be characterized based on three key factors: the gene to be transferred; the target tissue into which the gene will be introduced; and the gene delivery vehicle (vector) employed. Although the gene is the active therapeutic agent, the vector, which is usually derived from a virus, is a key determinant of both the toxicity and therapeutic success of the product. Two types of viral vectors are used to achieve long-term expression of the transferred gene: integrating vectors (typically lentiviral), which introduce the gene into a stem cell ex vivo and allow the donated gene to be passed to every daughter cell; and non-integrating vectors (typically adeno-associated), which introduce the gene in vivo into a long-lived postmitotic cell, such as a neuron, resulting in long-term expression provided that the donated DNA can be stabilized in the cell. In this review we delve into the state of the arts of in vivo and ex vivo gene therapy-based approaches that target the CNS to treat neuronopathic MPS, review the corresponding clinical trials conducted to date, and discuss the future implications of these therapies.

## 2. In Vivo Gene Therapy for Neuronopathic MPS

### 2.1. Adeno-Associated Virus (AAV)

AAV is a small (25 nm) non-enveloped parvovirus composed of an icosahedral capsid containing a single-stranded 4–7 Kb DNA genome. AAVs replicate only in the presence of helper viruses such us adenoviruses, herpesviruses, and papillomaviruses [27]. Since their discovery in the 1960s [28], AAVs have been successfully leveraged for clinical use. Glybera^®^ (alipogene tiparvovec) became the first-ever commercially available gene therapy [29], followed by Luxturna^®^ (voretigene neparvovec) and Zolgensma^®^ (onasemnogene abeparvovec). Due to their high efficiency, favorable safety profile, and non-cytotoxic characteristics, AAVs are among the most widely employed viral vectors in gene therapy.

Natural exposure to wild-type AAVs in the human population usually occurs during childhood (1–3 years of age) and does not cause any known illness. After infection, AAVs remain latent, requiring the coinfection of a helper virus to allow their replication [30]. At least 12 naturally occurring serotypes, which vary in their tissue tropism, have been discovered and classified to date [31]. Tissue tropism is used to target AAV gene-therapy vectors to tissues or organs of interest. AAV serotypes 1, 2, 5, 8, and 9 and recombinant human (rh)10 are the most studied CNS-targeting AAVs. These serotypes efficiently transduce neurons. Their ability to transduce other cells such as astrocytes, oligodendrocytes, and microglia is limited, but can be improved through the use of cell-specific promoters. In recombinant AAV (rAAV) vectors, different capsids exert different transduction effects and can also have distinct immunological properties [32].

### 2.2. Humoral and Cellular Immunity to AAV Vectors

After publication of initial evidence indicating that AAV vectors can elicit capsid T-cell responses in humans and thereby affect the duration of transgene expression [33,34], it became clear that vector immunogenicity is of the utmost importance for their application in the clinical field. The use of AAV vectors continues to pose several challenges related to vector interactions with the host immune system and the consequent impact on multiple treatment-related parameters, including transgene expression, durability, and toxicity, and efficacy, loss of which may entail re-administration of the vector [35,36].

Neutralizing antibodies (Nabs) against AAV varies geographically and are found in large proportions (30–60%) of the human population [37,38]. Consecutive infections may explain the high prevalence of anti-AAV Nabs, which results in broad cross-reactivity across different serotypes [37]. The production of Nabs of all four IgG subclasses has been reported. Preexisting immunity to AAVs is a key challenge in gene therapy, resulting in the exclusion of patients from clinical trials. A 2017 study of AAV seroprevalence rates in a population of MPS III (A and B) patients reported that while rates of seropositivity in MPS III patients were similar to those of age-matched healthy controls, in children aged less than 8 years, AAV IgGs for the majority of serotypes, especially those against AAV1 and AAVrh74, were more prevalent in MPS III patients than controls, suggesting that comorbid diseases and the patient’s environment may increase exposure to AAV infections [38].

Adaptive immune responses to AAV exposure include mediated and cellular immunity, both related to B- and T-cell activation [39]. Accordingly, patients that undergo gene therapy with AAV vectors develop anti-AAV IgG and IgM antibodies, resulting in high neutralizing titers that likely prevent vector re-administration. Recent research has also reported that complement proteins can bind to AAV capsid acting as mediators of vector immunogenicity, thus directly interacting with the innate immune system [40]. To address this issue, different approaches have been used to regulate the immunological response to AAV vectors. Administration of immunomodulatory drugs or depletion of B-cells prior to gene transfer have proven effective in blocking humoral immune responses to AAV vectors in preclinical and clinical studies [41,42]. Rituximab combined with rapamycin is currently being tested in patients with late-onset Pompe disease as a strategy to enable vector redosing (Clinical trials.gov: NCT02240407). An IgG-cleaving endopeptidase called imlifidase (IdeS) was recently evaluated as a means of removing anti-AAV IgGs and Nabs, and was shown to restore hepatic transduction both in vitro and after IV administration of AAV gene therapy in mice and non-human primates (NHPs) [43]. These results suggest that IgG cleavage by IdeS may overcome the effect of neutralizing antibodies in patients with preexisting anti-AAV antibodies and potentially enable vector re-administration.

### 2.3. Routes of Administration AAV Vectors Targeting the CNS

Several strategies to deliver AAVs to target the CNS in neuronopathic MPS have been developed and tested in animal models and in clinical trials. The two main routes of administration are systemic administration and direct injection into the CNS, each of which have specific benefits and potential risks [44,45]. The main clinical trials of gene therapies for neuronopathic MPS are listed in Table 1.

#### 2.3.1. Direct Delivery into the Central Nervous System

CNS-directed delivery is a compelling approach as lower vector doses are required to achieve clinically relevant transgene expression in CNS tissues, thereby reducing the overall immune response. The main strategies for CNS-directed AAV delivery researched to date include intracerebroventricular (ICV), intracisternal (IC), intrathecal (IT), and intraparenchymal administration (Figure 2). However, these CNS-directed delivery approaches are insufficient to completely block the effect of circulating anti-AAV antibodies and the administration procedures themselves can physically disrupt the BBB, providing even greater access to the brain to circulating antibodies that can then neutralize the vector [45]. It is also important to bear in mind that exposure to high systemic doses of AAV induces peripheral immune responses, and to therefore consider the possibility that neuroinflammation could also inform AAV immunogenicity in the CNS.

Intraparenchymal administration provides high target specificity, and therefore allows for administration of lower doses of experimental product. However, it requires an invasive neurosurgical technique guided by intraoperative radiological monitoring [46]. Furthermore, compared with systemic or CSF administration, direct delivery of relatively lower doses into an immune-privileged site such as the brain also reduces the effect of pre-existing antibodies against AAV serotypes. Studies have been performed using different sites of injection into the brain (cortex, striatum, hippocampus, thalamus, ventral tegument, cerebellum) and different AAV serotypes (1, 2, 5, 9, rh8 and rh10) in both small and large animal models of neuropathic lysosomal storage diseases (LSDs), including MPS I, MPS III, and MPS VII, with promising findings [47,48,49,50,51]. This approach has proven effective in reversing primary and secondary GAG accumulation and improving the behavioral phenotype and neuropathological parameters [52]. The main disadvantages of this approach include the highly invasive surgical procedure required and the limited distribution of the lysosomal enzymes from the injection site.

Direct delivery of AAV into the CSF offers another means of targeting the CNS. This approach has been used successfully to ameliorate certain phenotypic traits in small and large animal models of LSDs, including MPS I, MPS II, MPS III, and MPS VII, using several different AAV serotypes (1, 2, 4, 5, 8, 9, rh8, and rh10) administered into the ventricles, cisterna magna, or spinal cord [53,54,55,56]. Of the AAV serotypes tested, serotype 9 has proved the most promising. By targeting the ventricles or subarachnoid spaces through which CSF flows, the experimental product can reach all CNS structures that are contacted by the CSF. Somatic tissues can also be targeted through leakage of the vector into the bloodstream in the absence of serum antibodies against the vector. Intrathecal (IT) administration via lumbar puncture is likely the least invasive method for CSF delivery, and results in robust transduction across the CNS, including spinal neurons, sensory neurons, and dorsal root ganglia (DRG) at every level of the spinal cord, but provides limited distribution to the brain [57]. ICV administration requires specific neurosurgical procedures that have been evaluated in different clinical trials and has the advantage of providing broad distribution of the vector throughout the CNS. However, it is an invasive approach in which the needle tract crosses the brain parenchyma, risking injury and exacerbation of the immune response. Compared with ICV administration, IC delivery via the cisterna magna allows more directed targeting of the cerebellum, brainstem, and spinal cord [58]. However, in contrast to intraparenchymal and intraventricular delivery, there are no established protocols for the surgical approach, and, despite negligible, the potential risk of spinal injury cannot be completely ruled out [45].

#### 2.3.2. Systemic Delivery

Specific AAV serotypes (e.g., AAV9, AAVrh10, AAVrh8) that can cross the BBB and target the CNS [59,60] are administered intravenously (IV) as AAV-mediated gene therapy to treat neuronopathic MPS. This strategy has been employed in mice and large animal models of MPS I [61], MPS II [20], MPS III [19,62], and MPS VII [63]. AAV9 is probably the most promising intravenously delivered AAV that targets the CNS. Zolgensma^®^ is the only CNS-targeting intravenously administered AAV gene therapy approved by the FDA (since May 2019) for the treatment of SMA patients younger than 2 years of age, and proof that AAV9 efficiently crosses the BBB and transduces neurons. This therapy is administered as a single IV dose of 1.1 × 10^14^ vg/kg and results in longer survival, improved achievement of motor milestones, and better motor function compared with historical cohorts [64]. IV administration is noninvasive and allows the AAV to reach peripheral organs, thereby treating some of the somatic manifestations of MPS, although a higher vector copy number per kg body weight is required compared with direct administration into the CNS. Despite the advantages of systemic AAV administration to treat CNS diseases, application of this approach is hindered by several issues, including the efficiency with which the AAV crosses the BBB, peripheral toxicity, and immune elimination [65].

### 2.4. Toxicity of AAV Vectors

In recent years, different toxic effects have been attributed to AAVs (related to the capsid, the transgene product, or the immune response to both capsid and transgene) in animal models and humans, mainly after administration of high vector doses. In 2021, Kuzmin et al. [66] analyzed all AAV clinical trials conducted in Europe and the United States up to January 2020. The review included 94 trials that recruited a total of 3328 patients who received AAV gene therapies administered via different routes. No studies were terminated due to safety-related issues and 21% of studies reported administration-related serious adverse events (SAEs). Intravenous and intrathecal were considered the safest routes of administration. A similar percentage of SAEs was observed for intracranial administration, although these events tended to be of a higher grade and more clinically significant.

#### 2.4.1. Toxicity Related to CNS Delivery

In 2020, Hordeaux et al. [67] reported that sensory neuron degeneration in the dorsal root ganglia (DRG) was almost universal in NHPs in which AAVs were administered via the CSF, although the vast majority exhibited no clinical signs. DRG degeneration was also observed in humans in 2020, when one patient who received a lumbar injection of AAVrh10-miR-SOD1 developed a tingling sensation in the hands 3 weeks post-infusion, followed by pain and loss of strength in the left foot. A decrease or loss of SNAP amplitudes was observed in multiple nerves, suggesting neuronal loss in the DRG. Data indicating DRG toxicity and secondary axonopathy in NHPs suggest overexpression-mediated toxicity and apoptosis of sensory neurons that are highly transduced. In October 2020, Lysogene reported the death of a MPS IIIA patient recruited in the AAVance trial, although the cause of the death has not been indicated. White matter abnormalities near the injection site have also been observed in the AAVance trial, albeit without associated clinically significant symptoms. These effects, which may be related to neuroinflammation, are still being investigated.

#### 2.4.2. Toxicity Related to Systemic Delivery

A 2018 preclinical study by Wilson et al. [68] reported serious adverse events leading to death of some NHPs and piglets that received high doses of a systemic AAV9 variant (10^14^ GC/kg range). NHPs presented thrombocytopenia, coagulopathy, and increases in liver enzyme levels. Piglets demonstrated no evidence of hepatic toxicity, but within 14 days of vector injection, all three animals exhibited proprioceptive deficits and ataxia. In 2018–2019 multiple trials in which high doses of AAV9 (3 × 10^13^–3 × 10^14^ vg/kg) were administered reported serious adverse events leading to clinical holds, including thrombocytopenia, complement activation, and kidney damage. Administration of steroids and eculizumab (a complement C5 inhibitor intended for patients with atypical hemolytic uremic syndrome) in severe cases led to complete recovery. In 2020, progressive fatal liver toxicity was observed in three patients with X-linked myotubular myopathy who received a 3 × 10^14^ vg/kg dose of AAV8 [69]. The cascade of events that lead to intravascular coagulation and liver damage is not fully understood, and may involve alternative complement activation and a CD8+ T-cell response to the capsid, respectively. Hepatotoxicity following administration of Zolgensma^®^ for the treatment of SMA has also been described but patients recovered completed with no lethal outcomes [70].

### 2.5. Clinical Trials of AAV Vectors to Treat Neuronopathic MPS

#### 2.5.1. The Following Is a List of Clinical Trials of Gene Therapy Approaches Involving CNS Delivery of AAV Vectors to Treat Neuronopathic MPS

MPS I (Hurler syndrome).

A biotechnology company (Regenxbio) is conducting a phase I/II gene therapy trial with an intracisternally administered AAV9 vector containing an α-L-iduronidase (IDUA) expression cassette (RGX-111) (ClinicalTrials.gov Identifier: NCT03580083). The trial will recruit up to five patients aged four months and older into two cohorts, which will receive doses of either 1 × 10^10^ vg/g brain mass (cohort 1) or 5 × 10^10^ vg/g brain mass (cohort 2). Patients will receive immunosuppressive treatment for 48 weeks. In May 2021 Regenxbio completed dosing of patients in cohort 1. To date, no interim results have been published.

MPS II (Hunter syndrome).

Regenxbio is also conducting a phase I/II gene therapy trial for patients with neuropathic MPS II using an AAV vector that is directly delivered into the CNS (ClinicalTrials.gov Identifier: NCT04571970). RGX-121 is a recombinant AAV9 capsid containing a human iduronate-2-sulfatase expression cassette. This trial recruited patients with neuropathic Hunter syndrome (age, 4 months to 5 years) into three cohorts that receive the following doses: 1.3 × 10^10^, 6.5 × 10^10^; or 2.0 × 10 vg/g brain mass. The experimental product is administered intracisternally guided by computed tomography and patients receive immunosuppressive treatment for 48 weeks. To date, nine patients have been recruited and RGX-121 has been well-tolerated. In all five patients in cohort 2, reductions in CSF levels of HS and D2S6 have been observed up to 2 years after treatment, together with measurable CSF levels of iduronate-2-sulfate, which were undetectable in all patients prior to treatment. Combined median reductions of HS from baseline were 30.3% at week 8 and 35.0% at the last timepoint available for each patient. Patients also showed decreased levels of D2S6, a component of HS, up to 2 years after RGX-121 administration, with median reductions from baseline of 44.1% at week 8 and 40.4% at the last timepoint available for each patient. Furthermore, CSF levels of I2S enzyme, which were undetectable in all patients prior to treatment, were measurable in all five patients from cohort 2 after RGX-121 administration. Assessment of cognitive development in four of five patients revealed continued acquisition of language and/or motor skills in all patients followed up for >6 months [71,72].

MPS III (Sanfilippo syndrome).

The AAVance clinical trial is a phase II/III gene therapy trial for patients with MPS IIIA conducted by Lysogene (ClinicalTrials.gov Identifier: NCT03612869). The experimental treatment involves direct injections of AAV rh10 containing the human N-sulfoglucosamine sulfohydrolase cDNA (LYS-SAF 302) into both sides of the brain via image-guided tracks. The trial recruited children diagnosed with MPS IIIA, provided they were aged over 6 months and had a developmental quotient (DQ) ≥50%. Immune suppression was achieved by administration of tacrolimus, mycophenolate mofetil, and steroids. To date 19 children have been treated, with age at treatment ranging from 10 to 65 months. One death (a 43-month-old girl) was reported 18 months after gene therapy administration, although Lysogene reported that there was no evidence implicating the gene therapy procedure in the cause of death. White matter abnormalities near the injection sites have been detected, and the AAVance trial has been paused by the FDA since June 2020. No clinically relevant symptoms were attributed to these white matter abnormalities. Lysogene is collecting additional data to better characterize the safety profile of the experimental product. LYS-SAF 302 resulted in 30% and 37% reductions in CSF levels of HS and GM3, respectively, at 12 months (five patients completed 1 year of follow-up). An efficacy analysis of neurocognitive outcomes will be conducted on completion of 2 years of follow-up [73].

A phase I/II clinical trial of gene therapy based on AAV9 injection into the lateral ventricles of MPS IIIA patients is being conducted by ESTEVE and the Universitat Autónoma de Barcelona. In this study, three patient cohorts receive a single ICV dose of AAV9-hSGSH vector (cohort 1, 6.8 × 10^13^; cohort 2, 1.4 × 10^14^; cohort 3, 2.1 × 10^14^ vg/patient) into the cisterna magna. MPS IIIA patients aged 2–5 years are eligible for recruitment, and receive no concomitant immunosuppressant treatment. To date, nine patients have been treated (cohort, 1 *n* = 1; cohort 2, *n* = 3; cohort 3, *n* = 3) with a single neurosurgical intervention, and no significant incidents have been reported during the surgical procedure. Humoral immune responses were observed as expected (increase in total antibodies and Nabs against AAV9 in serum and CSF, but not against SGSH). Cellular immune responses, when detected, were transient, of low intensity, and without clinical manifestations. Only one serious adverse event was reported: a cerebrospinal fistula related to the ICV procedure in one patient [74]. To date no data regarding biomarkers and neurocognitive outcomes are available.

#### 2.5.2. Clinical Trials of Gene Therapy Approaches Involving Systemic Delivery of AAV Vectors to Treat Neuronopathic MPS Are Listed Below

MPS III (Sanfilippo syndrome).

Transpher A (ClinicalTrials.gov Identifier: NCT02716246) is a phase I/II clinical trial sponsored by Abeona Therapeutics that is currently recruiting MPS IIIA patients ranging in age from newborn to 2 years of age (or those older with a DQ ≥60). A self-complementary AAV9 carrying the human SGSH gene under the control of a U1a promoter (ABO-102) is delivered in a single dose via a venous catheter inserted into a peripheral limb vein. A tapering course of prophylactic enteral prednisone or prednisolone is also administered for a period of at least 2 months. To date 3 patients have been recruited in cohort 1 (dose 5 × 10^12^ vg/kg), 3 patients in cohort 2 (1 × 10^11^ vg/kg), and 15 patients in cohort 3 (dose 3 × 10^13^ vg/kg), with a mean follow-up duration of 32.6 months in cohort 3 to date. The experimental product has been well-tolerated with no serious adverse events. ABO-102 induces significant, sustained, dose-related reductions in disease-specific biomarkers 2 years post-administration: CSF levels of HS decreased to the lower limit of quantitation (70–75% in cohort 3) and CSF GM2 and GM3 levels decreased in treated patients in cohort 3. Moreover, a sustained decrease in liver volume has been observed in all cohorts. Assessment of neurocognitive outcomes has shown that in cohort 3, treated children aged <2 years or with a DQ ≥60 showed continuous developmental progress 30–36 months post-administration (43, 48, and 64 months of chronological age), a time at which they should be experiencing cognitive decline according to natural history data. One child treated at 1 year of age continues to track on the DQ100 line 2.5 years after treatment, showing normal development [75,76].

Transpher B is a phase I/II clinical trial also sponsored by Abeona Therapeutics and currently recruiting patients with MPS IIIB aged 0–2 years (or older with a DQ ≥60). A single dose of a self-complementary AAV9 carrying the NAGLU gene under the control of a CMV enhancer/promoter (rAAV9.CMV.hNAGLU) (ABO-101) is administered intravenously into a peripheral limb vein. A tapering course of prophylactic enteral prednisone or prednisolone is also administered for a period of at least 2 months. To date two patients have been recruited in cohort 1 (dose, 2 × 10^13^ vg/kg), five patients in cohort 2 (5 × 10^13^ vg/kg), and four patients in cohort 3 (dose 3 × 10^14^ vg/kg), with a mean follow-up duration of 12.4 months in cohort 3 to date. As in the Transpher A trial, the experimental product has been well-tolerated with no serious adverse events. Treatment with ABO-101 is associated with a sustained, dose-dependent improvement in CNS and systemic biomarkers indicating a potent biologic effect post-treatment. Specifically, CSF HS levels decreased to the lower limit of quantitation (80% in cohort 3), an effect that persisted for up to 24 months, and dose-dependent normalization of plasma NAGLU activity up to month 6 was observed in cohort 3. A sustained decrease in liver volume has also been observed. No effect on cognitive outcomes has been described in cohort 1. Cognitive evaluation will require longer follow-up of treated children from cohorts 2 and 3 [76,77].

## 3. Ex Vivo Gene Therapy for Neuronopathic MPS

Ex vivo gene therapy is a second approach that has shown promise for the treatment of neurological MPS. The objective is to extract cells from the patient, modify them genetically and transplant them back as other HSCT. The new cells will secrete enzymes that will be able to be captured by deficient cells. The most common cellular lines used in LSD therapies are hematopoietic precursor cells (HPCs) which can repopulate the fixed tissue macrophage system and directly deliver enzymes to multiple organs.

Initial ex vivo HSC-directed gene therapy with murine retroviral vectors studies were performed in mouse models of LSD and showed favorable results with significant improvement in systemic symptoms. More studies were performed using vectors containing promoters to facilitate direct cell-specific expression of enzymes. The combination of HPC with an erythrocyte promoter has also proven effective for the treatment of galactosialidosis. With all these promising results hematopoietic-directed gene therapy has been considered to probably be effective for the treatment of several LSDs.

### 3.1. Cross Correction Mechanism

The mechanism of cross correction forms the rationale for both ex vivo gene therapy and HSCT. Cross correction was discovered in 1968 by Neufeld et al., who described the correction of biochemical defects in skin fibroblasts from MPS I and II patients when cultured in vitro with each other or with healthy cells [78]. Some years later it was shown that a functional enzyme secreted by wild-type cells is taken up into the lysosomal compartment through the mannose-6-phosphate receptor (M6P-R) pathway [79]. In the synthetic pathway, M6P-R is added to the enzyme in the Golgi apparatus, and both are then transferred to late endosomes and lysosomes. About 40% of the enzyme is secreted into the extracellular space and can be captured via endocytosis by the surrounding cells following binding of M6P-R to the cell membrane [80]. This mechanism is used as the basis for HSCT in some neurodegenerative LSDs in which transplanted cells can reach the CNS via macrophages and microglia and serve as permanent sources of functional lysosomal enzymes after a single procedure, provided that complete engraftment occurs. For MPS IH, HSCT at young ages is the best therapeutic strategy, although some tissues such as bone, cardiac valves, and CNS manifestations do not always present an optimal evolution. Despite its efficacy, allogeneic HSCT can have severe side effects due to the conditioning regimens required, as well as graft versus host disease (GvHD). Moreover, it can take considerable time to identify suitable donors, further delaying the procedure. In the CNS, the effects of HSCT can begin 10 to 12 months after infusion, during which time neurological regression can occur. Multiple factors, predominantly age, clinical status, and donor type, can affect the final treatment outcome.

Ex vivo gene therapy combines autologous HPCs and gene therapy vectors in order to increase enzyme production by the transplanted cells while reducing morbidity [81]. On one hand, genetically modified cells can produce higher than normal enzyme concentrations, thereby increasing the enzyme dose delivered to the intracellular space of the affected tissues. On the other hand, the use of milder conditioning regimens and the avoidance of GvHD can reduce morbidity and mortality associated with the transplant procedure.

### 3.2. Lentiviral Vectors

The benefits of HSCT for the treatment of certain lysosomal disorders have been widely reported. Therapeutic effects of HSC gene therapy were first proven in animal models without neurological involvement that were treated with gamma retroviral vectors (GRVs), which increased enzyme concentrations and reduced substrate accumulation, with consequent positive clinical effect. Important advances were made by employing lentiviral vectors (LVs), which improved transduction efficiency and gene expression in grafted cells. HSCT using LV-transduced cells carrying the normal cDNA of the gene was shown to restore lysosomal enzyme activity to above normal values in the hematopoietic system and affected tissues, including the brain, resulting in marked amelioration of the disease phenotype [82].

The potential of LV-HSCT to treat neurological disease was first demonstrated in a mouse model of metachromatic leukodystrophy (MLD) [83]. Previous studies using GRVs succeeded in treating somatic symptoms but not neurological regression. The use of LVs in presymptomatic or oligosymptomatic animals effectively prevented the development of neurological alterations in both the CNS and the peripheral nervous system and corrected somatic manifestations. The benefits of LV-HSCT have been confirmed in other studies of neurological LSD including GM1 gangliosidosis, globoid cell leukodystrophy, and MPS IH and IIIA [84,85]. Overexpression of the enzyme in macrophages and microglia appears to help reverse the pathological cascade and prevent both inflammatory responses and neurodegeneration.

Lentiviral vectors are retroviruses with a single-stranded RNA genome that can reverse transcribe RNA into DNA. Viruses enter the cell by endocytosis or fusion after bonding to the membrane. Once inside they are able to release the core together with a reverse transcriptase which is able to synthesize cDNA from viral RNA and transport the cDNA to the nucleus. This process requires the utilization of the cell’s nucleotides and ends with the integration of the new DNA into the host genome. Unlike other retroviruses, LVs can infect cells regardless of whether they divide: this explains their targeting of CNS cells, which usually do not divide. Lentiviral particles are divided into ‘‘generations’’ according to the packaging plasmid used for production. Each modification leads to safer and more efficient generation. In the most recent gene therapy studies, third generation LVs are used [86,87,88,89].

LVs are not exempt from safety issues, which include insertional mutagenesis and persistent expression of site-specific nucleases leading to off-target mutations.

### 3.3. Ex Vivo Gene Therapy for the Treatment of Neuropathic MPS

The ability of HSC gene therapy based on LVs to correct neurologic and skeletal defects in MPS IH has been investigated in mice, in which its efficacy on bone involvement surpasses that of HCT alone and is dependent on achieving higher than normal levels of enzyme activity in the different tissues [90]. The amount of enzyme delivered to the CNS can change depending on the expression of the promoter which is used: a ubiquitous promoter will be more efficient than an erythroid- or platelet-specific promoter. It is for this reason that treated myeloid able to overexpress the enzyme will possibly increase significantly the delivery of IDUA to the CNS and ideally also to bone. Sergijenko and colleagues [91] demonstrated correction of neurologic disease in a mouse model of MPS IIIA using a myeloid lineage-specific promoter with expression of N-sulfoglucosamine sulfohydrolase enzyme cDNA. Wakabayashi et al. [84] reported positive effects on CNS lesions in a mouse model of MPS II after HSC gene therapy, with the correction of neuronal manifestations by ameliorating lysosomal storage and autophagic dysfunction. Taken together, the results of these studies indicate that higher than normal levels of enzyme expression induced by HPC gene therapy could be therapeutically beneficial in patients who tend to respond poorly to allogenic HCT.

The main determinants of the efficacy of HSC gene therapy are whether higher than normal enzyme expression in brain cells is achieved, the pre-transplant conditioning regimen used, and, as applies to all therapies, timeliness (the earlier the better). Conditioning regimens usually involve lower doses than for allogenic HCT in order to reduce toxicity in cases in which there is no CNS involvement. In the case of neuronopathic LSDs, in which high levels of engraftment and reproduction of myeloid cells in the brain and microglia are required, specific regimens (e.g., busulfan) and higher doses are used in order to promote myeloid cell turnover in the brain [92,93].

### 3.4. Clinical Trials of Ex Vivo Gene Therapy

MPS I: Orchard Therapeutics is leading a trial to evaluate the safety and efficacy of OTL-203 (ClinicalTrials.gov Identifier: NCT03488394), an investigational ex vivo autologous HSC gene therapy for patients with Hurler syndrome, conducted at the San Raffaele Telethon Institute for Gene Therapy (SR-Tiget) in Milan, Italy (18). Eight patients have been treated with genetically modified autologous HSCs collected from mobilized peripheral blood (or bone marrow) and transduced with IDUA lentiviral vector encoding the human IDUA gene. All patients are diagnosed with MPS-IH and are ages from 1 month to 11 years. Initial results reported by the company indicate that the treatment is generally well-tolerated with a safety profile consistent with the selected conditioning regimen. All patients achieved supra-physiological IDUA expression in dried blood spot samples at 12 months which is a primary efficacy endpoint. Furthermore, increased IDUA expression was detected in cerebrospinal fluid (CSF), together with a reduction of GAG levels in CSF and normalization of GAG levels in urine. All patients have a follow-up ranging from 6 months to 2 years in which cognitive evaluations using the Bayley scale maintained in normal ranges. Height growth was also within age-appropriate reference ranges for all patients post-treatment, with follow-up ranging from 9 months to 2 years. In the same way, all patients showed stabilization of motor function and improved range of motion and anti-IDUA antibodies were not detected in any patient within 2 months of treatment.

MPS II: Avrobio is expecting to begin a phase 1/2 trial with lentiviral gene therapy for MPS II at the University of Manchester in the second half of 2021. Published preclinical data demonstrate that the introduction of the transgene has the potential to correct peripheral disease and normalize brain pathology [94]. The investigational gene therapy involves ex vivo transduction of the patient’s own hematopoietic stem cells.

MPS IIIA: Orchard Therapeutics is also leading a trial to evaluate the safety and efficacy of OTL-201, an investigational ex vivo autologous HSC gene therapy for patients with Sanfilippo syndrome type A (ClinicalTrials.gov Identifier: NCT04201405). The trial is conducted at the San Raffaele Telethon Institute for Gene Therapy (SR-Tiget) in Milan, Italy. The results reported for the first three patients in a 3-month follow-up period show that it has been generally well-tolerated. There were some transplant-related SAEs and adverse events which were resolved. The laboratory data indicate that hematological engraftment is achieved with a rapid recovery of neutrophil, platelet, and hemoglobin levels, and in the same period SGSH enzyme expression in leukocytes and CD15+ cells increased from undetectable at baseline to supra-physiological levels at 3 months. Urine GAG levels decreased to within the normal range by 3 months post-treatment.

## 4. Gene Editing for Neuronopathic MPS

Direct repair of defective genes by site-specific in vivo genome editing is another important area of investigation within the field of in vivo gene therapy for genetic diseases [95]. Several genome editing technologies have been developed over the past 10 years [96]. The most promising genome editing tools are those based on programmable nucleases, which include the clustered regularly interspaced short palindromic repeat (CRISPR)–Cas9 system, zinc finger nucleases (ZFNs) and transcription activator-like effector nucleases (TALENs). Different in vivo and ex vivo genome editing platforms have been shown to effectively target the CNS in preclinical studies of neuronopathic MPS, including MPS I [97] and MPS II [98]. Currently, targeted insertion of the functional gene at the albumin locus in hepatocytes through in vivo ZFN-mediated genome editing is being studied as a potential therapy for attenuated forms of MPS I and MPS II [99]. The first in vivo human genome editing trial was conducted by Sangamo Therapeutics in 2018 [100] in patients with an attenuated form of MPS II. This multicenter phase I/II clinical trial (CHAMPIONS) employed ascending intravenously administered doses of SB-913, which contains a zinc finger nuclease system, the correct IDS gene, and an AAV type 2/6 vector. Corticosteroids were administered for immune suppression. The study included nine patients aged 5 years and older, divided into 3 cohorts, each of which received a different dose (5 × 10^12^ vg/kg, 1 × 10^13^ vg/kg, and 5 × 10^13^ vg/kg). Regarding the question of whether the IDS gene had been integrated, Sangamo reported mixed results. In the low-dose group, no evidence of gene integration was obtained for one patient, while another was unevaluable as they were unable to undergo liver biopsy. The two patients who received the medium dose showed evidence of gene integration, but no increase in plasma IDS activity. In the high-dose group, one patient was unable to undergo biopsy to assess gene integration, and results from the other patient in this group (Patient 6) are pending. Patient 6 was the only patient to show an increase in plasma IDS, although this increase disappeared after increases in liver enzyme levels were detected. No measurable decrease in urine GAG levels was detected in any of the patients. Sangamo scientists developed a specific assay to detect gene integration by identifying albumin-IDS chimeric mRNA transcripts using reverse transcription polymerase chain reaction (RT-qPCR). Sangamo is also conducting a phase I/II clinical trial of ZFNs for MPS I (EMPOWERS), in which three different cohorts receive ascending doses of the genome editing components (ZFN1, ZFN2, and IDUA donor). Preliminary results [101] suggest that the treatment is safe although its efficacy remains to be proven. Measurement of plasma IDUA activity has shown no significant changes from pre-treatment values.

## 5. Conclusions

Gene therapies for neuronopathic MPS are rapidly advancing. Preclinical studies have demonstrated profound therapeutic benefit in animal models prompting the development of clinical trials targeting the brain. There are two types of gene therapy approaches for CNS involvement: the direct transfer of a therapeutic gene into brain cells and HSC-targeted gene therapy. AAVs are the most popular viral vectors in gene therapy for MPS to date, followed by ex vivo gene therapy with lentiviral vectors. Despite encouraging results, several challenges remain to be overcome in clinical settings, including vector selection, administration route, and immunogenicity.

Gene editing technology has also been recently applied in this field, with two trials focused on MPS I and MPS II. Although still not proven to be effective in humans, the fact that in vivo genome editing was first-in-humans for MPS is extremely promising.

## Figures and Tables

**Figure 1 ijms-22-09200-f001:**
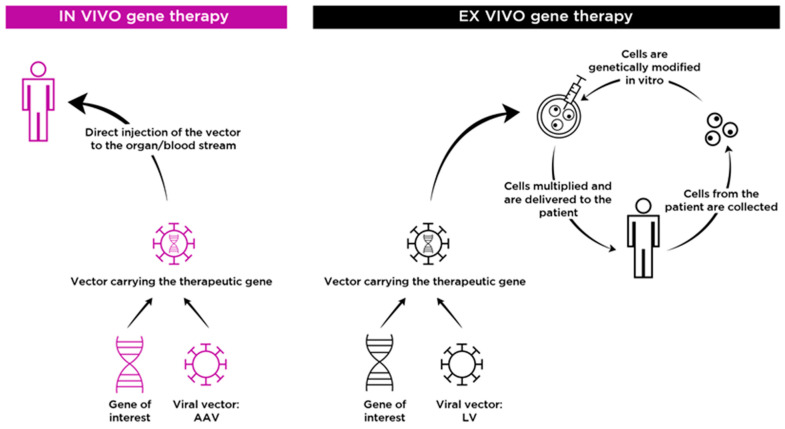
In vivo and ex vivo gene therapy approaches for MPS. In in vivo gene therapy the vector carrying the therapeutic gene is directly delivered to the target organ/blood stream of the patient. In ex vivo GT approaches, patient cells are collected and stem cells are genetically modified in vitro and the final transduced stem cells are delivered to the patient after administration of a conditioning regimen.

**Figure 2 ijms-22-09200-f002:**
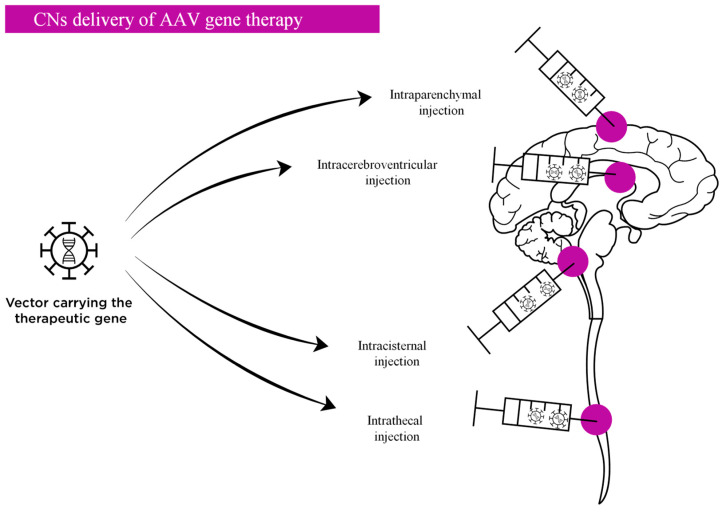
In vivo direct CNS administration routes for neuronopathic MPS. This approach includes intraparenchymal, intracerebroventricular (ICV), intracisternal (IC) and intrathecal (IT) administration. Direct CNS-targeting strategies generally represent highly invasive approaches for human therapeutic application.

**Table 1 ijms-22-09200-t001:** Clinical trials of ex vivo and in vivo gene therapy (GT) for neuronopathic MPS.

Disease	Study Phase	Experimental Product	Route	Sponsor
		in vivo gene therapy		
MPS I	I/II	AAV9 vector containing the human α-L-iduronidase gene	Intracisternal	Regenxbio
MPS II	I/II	AAV9 vector containing the human iduronate-2-sulfatase gene	Intracisternal	Regenxbio
MPS IIIA	II/III	AAV rh10 containing the human N-sulfoglucosamine sulfohydrolase gene	Intraparenchymal	Lysogene
MPS IIIA	I/II	AAV9 vector containing the human iduronate-2-sulfatase gene	Intracisternal	Esteve
MPS IIIA	I/II	AAV9 containing N-sulfoglucosamine sulfohydrolase expression cassette	Intravenous	Abeona
MPS IIIB	I/II	AAV9 containing the human α-N acetylglucosaminidase gene	Intravenous	Abeona
ex vivo gene therapy				
MPS I	I/II	Autologous CD34+ cells transduced with a lentiviral vector containing the human α-L-iduronidase gene	Intravenous	Fondazione Telethon
MPS II	I/II	Autologous CD34+ cells transduced with a lentiviral vector containing the human iduronate sufatase gene	Intravenous	Avrobio
MPS IIIA	I/II	Autologous CD34+ cells transduced with a lentiviral vector containing the human N-sulfoglucosamine sulfohydrolase gene	Intravenous	Orchard

## Data Availability

Not applicable.

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
