# Peer review of "Gene Therapy for Neuronopathic Mucopolysaccharidoses: State of the Art"

_ijms, 2021, doi:10.3390/ijms22179200_

Round 1

Reviewer 1 Report

This manuscript reviewed the update of gene therapy for neuropathic mucopolysaccharidoses. The authors made clear explanation on the therapy by dividing them into In Vivo gene therapy and Ex Vivo gene therapy. All paragraphs and figures were explained well. This is a nice review of gene therapy and may be shared by all physicians and scientists that investigate new treatment strategies for MPS.

Author Response

This manuscript reviewed the update of gene therapy for neuropathic mucopolysaccharidoses. The authors made clear explanation on the therapy by dividing them into In Vivo gene therapy and Ex Vivo gene therapy. All paragraphs and figures were explained well. This is a nice review of gene therapy and may be shared by all physicians and scientists that investigate new treatment strategies for MPS.

ANSWER: Thank you for assessing the manuscript and the positive comments.

Reviewer 2 Report

This review focuses on the state of the art of in vivo and ex vivo gene therapy-based approaches targeting the CNS in neuronopathic MPS, discusses clinical trials conducted to date, and provides a vision for the future implications of these therapies for the medical community.

Recent advances in the field, as well as limitations relating to efficacy, potential toxicity, and immunogenicity, are also discussed. AAV9 is probably the most promising intravenously delivered AAV that targets the CNS and the Systemic delivery and toxicity of AVV is also discussed. Finally, the gene editing for neuronopathic MPS has been extensively discussed.

Very interesting review in the field of the MPS treatments. Very easy to be read and very UpToDate review.

Only one concern:

Which are the sources of the data reviewed regarding the clinical trials???? Chapter 2.5. Clinical trials of AAV vectors to treat neuronopathic MPS. If possible, add references for the results of the clinical trials when published.

Author Response

 This review focuses on the state of the art of in vivo and ex vivo gene therapy-based approaches targeting the CNS in neuronopathic MPS, discusses clinical trials conducted to date, and provides a vision for the future implications of these therapies for the medical community. Recent advances in the field, as well as limitations relating to efficacy, potential toxicity, and immunogenicity, are also discussed. AAV9 is probably the most promising intravenously delivered AAV that targets the CNS and the Systemic delivery and toxicity of AVV is also discussed. Finally, the gene editing for neuronopathic MPS has been extensively discussed.

 Very interesting review in the field of the MPS treatments. Very easy to be read and very UpToDate review.

Only one concern:  Which are the sources of the data reviewed regarding the clinical trials???? Chapter 2.5. Clinical trials of AAV vectors to treat neuronopathic MPS. If possible, add references for the results of the clinical trials when published.

ANSWER: Thank you for the comments and efforts towards improving our manuscript. Regarding Chapter 2.5. International 2021 symposiums about the topic (16th International Symposium on MPS and Related Diseases and American Society of Gene and Cell Therapy's 24th Annual Meeting) were the sources for the most updated clinical trial results. The reference have been included in the manuscript. In addition, ClinicalTrials.gov Identifiers have been also added to the text when available.